# Enhancing antibody responses by multivalent antigen display on thymus-independent DNA origami scaffolds

Eike-Christian Wamhoff[1,11], Larance Ronsard [2,11], Jared Feldman[2,11], Grant A. Knappe [1,3,11], Blake M. Hauser [2,11], Anna Romanov [1,4], James Brett Case [5], Shilpa Sanapala[5], Evan C. Lam[2], Kerri J. St. Denis [2], Julie Boucau [2], Amy K. Barczak [2], Alejandro B. Balazs [2], Michael S. Diamond [5,6,7], Aaron G. Schmidt [2,8] ✉, Daniel Lingwood [2] ✉ & Mark Bathe [1,9,10] ✉

Protein-based virus-like particles (P-VLPs) are commonly used to spatially organize antigens and enhance humoral immunity through multivalent antigen display. However, P-VLPs are thymus-dependent antigens that are themselves immunogenic and can induce B cell responses that may neutralize the platform. Here, we investigate thymus-independent DNA origami as an alternative material for multivalent antigen display using the receptor binding domain (RBD) of the SARS-CoV-2 spike protein, the primary target of neutralizing antibody responses. Sequential immunization of mice with DNA-based VLPs (DNA-VLPs) elicits protective neutralizing antibodies to SARS-CoV-2 in a manner that depends on the valency of the antigen displayed and on T cell help. Importantly, the immune sera do not contain boosted, class-switched antibodies against the DNA scaffold, in contrast to P-VLPs that elicit strong B cell memory against both the target antigen and the scaffold. Thus, DNA-VLPs enhance target antigen immunogenicity without generating scaffold-directed immunity and thereby offer an important alternative material for particulate vaccine design.

Multivalent display of antigens on virus-like particles (VLPs) can improve the immunogenicity of subunit vaccines[1–3]. Nanoparticulate vaccines with diameters between 50 and 200 nm ensure efficient trafficking to secondary lymphoid organs, whereas particle diameters below 50 nm overcome undesired retention at the injection site and promote the penetration of B cell follicles[4,5]. In secondary lymphoid organs, multivalency promotes B cell receptor (BCR) crosslinking and signaling as well as

[1]Department of Biological Engineering, Massachusetts Institute of Technology, Cambridge, MA 02139, USA. [2]Ragon Institute of Massachusetts General Hospital, Massachusetts Institute of Technology and Harvard University, Cambridge, MA 02139, USA. [3]Department of Chemical Engineering, Massachusetts Institute of Technology, Cambridge, MA 02139, USA. [4]Koch Institute for Integrative Cancer Research, Massachusetts Institute of Technology, Cambridge, MA 02139, USA. [5]Department of Medicine, Washington University School of Medicine, St. Louis, MO 63110, USA. [6]Department of Molecular Microbiology, Washington University School of Medicine, St. Louis, MO 63110, USA. [7]Department of Pathology and Immunology, Washington University School of Medicine, St. Louis, MO 63110, USA. [8]Department of Microbiology, Harvard Medical School, Boston, MA 02115, USA. [9]Broad Institute of MIT and Harvard, Cambridge, MA 02139, USA. [10]Harvard Medical School Initiative for RNA Medicine, Harvard Medical School, Boston, MA 02115, USA. [11]These authors contributed equally: Eike-Christian Wamhoff, Larance Ronsard, Jared Feldman, Grant A. Knappe, Blake M. Hauser. ✉e-mail: aschmidt@crystal.harvard.edu; dlingwood@mgh.harvard.edu; mark.bathe@mit.edu

BCR-mediated antigen uptake, thereby driving B cell activation and humoral immunity[6–13].

The importance of BCR signaling for the generation of antibody responses was initially recognized for thymus-independent (TI) antigens, particularly of the TI-2 class[14–16]. The multivalent display of these non-protein antigens induces BCR crosslinking in the absence of T cell help. The resultant antibody responses proceed through extrafollicular B cell pathways, with limited germinal center (GC) reactions, affinity maturation, and induction of B cell memory[17,18]. Multivalent antigen display also enhances BCR-mediated responses to thymus-dependent (TD) antigens, namely proteins[8,9]. In this context, follicular T cell help enables GC reactions to generate affinity-matured B cell memory that can be boosted or recalled upon antigen reexposure[19–21]. Consequently, the nanoscale organization of antigens represents a well-established vaccine design principle not only for TI antigens, but also to elicit humoral immunity through the TD pathway[1–3].

Leveraging this design principle, protein-based virus-like particles (P-VLPs) have emerged as an important material platform for multivalent subunit vaccines[22–38]. P-VLPs enable the rigid display of TD antigens and have been used to investigate the impact of valency on B cell activation in vivo, suggesting early B cell activation and downstream humoral immune responses are improved for some antigens as valency increases[8–10]. However, control over antigen valency in P-VLPs is constrained to the constituent self-assembled protein scaffold subunits, rendering the investigation of antigen valency on humoral immunity challenging without simultaneously altering scaffold size, geometry, and protein composition[9,10]. Alternatively, if a constant protein scaffold geometry is used, then current approaches are limited to stochastically-controlled antigen valency and spatial positioning[8,29,30,38]. Furthermore, protein-based scaffolds themselves are TD antigens that elicit humoral immunity[38–40]. This potentially misdirects antibody responses from the target antigens of interest[41,42], and might also lead to imprinting[43] in which off-target, immunodominant epitopes distract from target epitopes of interest in generating de novo B cell memory. Finally, scaffold-directed immunological memory may also result in antibody-dependent clearance of the vaccine material, thereby limiting sequential or diversified immunizations with a given P-VLP[44,45].

We hypothesized that these limitations could be overcome by multivalent antigen display on a non-protein scaffold, which we could test by scaffolding a TD antigen on an icosahedral DNA origami nanoparticle that is a TI antigen. This platform provides unique access, compared with other materials, including proteins, to rationally designed DNA-based VLPs (DNA-VLPs) below the optimal 50 nm size-scale with scaffold-independent control over the valency and spatial organization of antigen display[46–51]. While we and others have leveraged these VLPs in vitro to probe the nanoscale parameters of IgM recognition[52] and BCR signaling in reporter B cell lines[53], the in vivo properties of this material remain largely unclear. Theoretically, the use of a TI scaffold could focus the antibody response on the target TD antigen of interest, while confining scaffold-directed B cell responses to the non-boostable TI pathway[54,55].

To test this hypothesis, we construct DNA-VLPs displaying the SARS-CoV-2 receptor binding domain (RBD) derived from the spike glycoprotein, a key target for eliciting neutralizing antibodies against this virus[56–59]. We find that sequential immunization with DNA-VLPs in mice boosts neutralizing and protective RBD-specific antibodies in a manner that is dependent on both antigen valency and T cell help. We further show that in contrast to P-VLPs, DNA-VLPs do not generate boostable antibodies against the scaffold. Collectively, this offers a proof-of-concept study that the antibody titer-enhancing benefits of multivalent protein antigen display can be decoupled from eliciting potent B cell responses against the scaffold platform itself.

## Results

### Design and fabrication of RBD-functionalized DNA-VLPs

SARS-CoV-2 trimeric spike glycoproteins are displayed on the surface of ~100 nm diameter virions[60], and each glycoprotein monomer contains the receptor binding domain (RBD) that engages the ACE2 receptor required for viral uptake. Because of this, the RBD is a key target for neutralizing antibody responses[56–59]. To ensure optimal trafficking of our vaccine platform, which requires particle diameters smaller than 50 nm[5], we computationally designed and fabricated an icosahedral DNA-VLP with 30 conjugation sites and a ~34 nm scaffold diameter to display the RBD[47,49]. A covalent post-assembly functionalization strategy employing strain-promoted azide-alkyne cycloaddition (SPAAC) chemistry was used to ensure rigid, irreversible antigen attachment to the DNA scaffold (Fig. 1a)[48], unlike previous work that used a reversible, hybridization strategy[53]. Towards this end, we synthesized 30 oligonucleotide staples bearing dibenzocyclooctyne (DBCO) groups at their 5′ ends to assemble DNA-VLPs symmetrically displaying 1, 6, or 30 DBCO groups on their exterior (Supplementary Fig. 1, Supplementary Tables 1–3). Employing a reoxidation strategy, the monomeric RBD was selectively modified at an engineered C-terminal Cys with a succinimidyl 4-(N-maleimidomethyl)cyclohexane-1-carboxylate (SMCC)-azide linker to yield RBD-Az, which was subsequently incubated with DBCO-bearing DNA origami to fabricate **DNA-VLP-1x, −6x**, and **−30x** (Fig. 1b, Supplementary Fig. 2, Supplementary Note 1). The optimization of reaction conditions yielded near-quantitative functionalization efficiency of conjugation sites as determined by denaturing, reversed-phase HPLC[48] and Trp fluorescence[53] (Fig. 1c and Supplementary Fig. 3). The monodispersity of purified DNA-VLPs was validated by dynamic light scattering (DLS) (Fig. 1d). Analysis of **DNA-VLP-30x** via negative-stain transmission electron microscopy (TEM) validated the structural integrity of the DNA origami with the designed size of ~40 nm (Fig. 1e and Supplementary Fig. 4). While the icosahedral geometry could not be fully resolved, presumably due to accumulation of uranyl formate in the interior of the DNA origami, antigens were clearly visible and organized symmetrically.

### DNA-VLPs are recognized by the ACE2 receptor

To investigate the binding activity of RBD-Az before and after conjugation to DNA-VLPs, we conducted flow cytometry experiments with ACE2-expressing HEK 293 cells (Fig. 2a). Initially, monovalent binding of wild-type monomeric RBD and fluorophore-labeled RBD-Cy5, obtained by selectively labeling the azide, was compared (Fig. 2b, c). The RBD constructs were incubated at 200 nM with the HEK 293 cells, and bound antigen was detected using CR3022[59], an anti-RBD antibody. These experiments revealed comparable binding between the two constructs, demonstrating preservation of binding activity of the receptor binding motif (RBM) in the RBD and the viability of the reoxidation strategy for selective labeling of the terminal Cys (Supplementary Fig. 2 and Supplementary Note 1).

Next, we explored whether multivalent RBD display using DNA-VLPs would result in increased avidity. Two additional fluorophore-labeled DNA-VLPs, **DNA-VLP-Cy5-30x** and **DNA-VLP-Cy5**, were fabricated to allow for direct detection of binding (Fig. 1b and Supplementary Fig. 1). Binding of **DNA-VLP-Cy5-30x** was enhanced compared to monomeric RBD-Cy5, whereas no binding was observed for the **DNA-VLP-Cy5** (Fig. 2d, e). When correcting for Cy5 brightness per RBD, **DNA-VLP-Cy5-30x** displayed an approximately ten-fold increase in median fluorescence intensity compared with monomeric RBD-Cy5, likely due to avidity effects of multivalent DNA-VLPs binding to the cognate ACE2 receptors.

## DNA-VLPs elicit valency-dependent B cell signaling in vitro

We then evaluated the impact of RBD-functionalized DNA-VLPs on BCR signaling using a previously described Ca²⁺ flux assay (Fig. 2a)[61]. Specifically, Ramos B cell lines expressing membrane anchored forms of the somatic CR3022 or B38 anti-RBD antibodies were established[59,62], and BCR signaling was validated by incubation with anti-IgM antibody. At 30 nM antigen concentration, monomeric RBD did not elicit B cell activation in vitro (Fig. 2f, g). In contrast, incubation of the Ramos B cells with multivalent DNA-VLPs at the same antigen concentration resulted in efficient BCR signaling. We further observed valency-dependent increases in total Ca²⁺ flux for both cell lines with **DNA-VLP-30x** showing greater potency than **DNA-VLP-6x**. CR3022 ($K_D \approx 0.27\,\mu M$, Fig. 2f) and B38 ($K_D \approx 1.00\,\mu M$, Fig. 2g) bind distinct RBD epitopes with moderate monovalent affinity as reported for the corresponding Fab fragments[39]. Despite this four-fold difference in affinity, we observed comparable total BCR signaling relative to the IgM control for all functionalized DNA-VLPs, consistent with previously described avidity effects at the B cell surface[63]. We conclude that our DNA-VLPs efficiently bound and induced signaling by RBD-specific BCRs in a valency-dependent manner, analogous to studies using similar assays to evaluate protein- and DNA-scaffolded multivalent subunit vaccines[53,61,64–70].

## DNA-VLPs elicit protective neutralizing antibody responses in a manner dependent on valency and T cell help

To investigate whether RBD-functionalized DNA-VLPs activated B cells in vivo to induce antibody responses, C57BL/6 mice were sequentially immunized intraperitoneally with monomeric RBD, **DNA-VLP-6x**, or **DNA-VLP-30x** at doses equivalent to 7.5 μg RBD with the Sigma

adjuvant system (Fig. 3a and Supplementary Fig. 3). Serum IgG responses against the RBD were monitored using ELISA and correlated with in vitro BCR signaling findings (Fig. 3b and Supplementary Fig. 5). Following boost 1, we observed an ~130-fold increase in endpoint dilutions for **DNA-VLP-30x** over monomeric RBD. In contrast, **DNA-VLP-6x** elicited comparable antibody responses with respect to monomeric RBD following each boost, suggesting that a higher minimum antigen copy number is needed to enhance B cell responses in vivo than in vitro (Fig. 2f, g), which may be due to differences in trafficking or degradation rates, for example, between **DNA-VLP-6x** and **DNA-VLP-30x**. Overall, IgG titers increased for monomeric RBD, **DNA-VLP-6x**, and **DNA-VLP-30x** following boost 2, converging to similar endpoint dilutions. These findings of earlier and stronger boosting of IgG titers and efficient B cell memory recall elicited by the **DNA-VLP-30x** are hallmarks of multivalent versus monomeric subunit vaccines[23,24], consistent with enhanced IgG titers elicited by P-VLPs of increasing valency[8–10].

To quantify the quality of antibody responses generated by DNA-VLPs, valency-dependent enhancement of RBD-specific antibody responses was interrogated using virus neutralization assays (Fig. 3c, d and Supplementary Fig. 6). In both pseudotype and authentic virus neutralization assays, serum derived from animals immunized with DNA-VLPs efficiently neutralized the ancestral Wuhan-1 strain of SARS-CoV-2. However, across both the pseudotype and authentic virus assays, **DNA-VLP-30x** elicited IgG titers with higher neutralization potency than both monomeric RBD and **DNA-VLP-6x**, indicating that the higher valency DNA-VLP induced superior quality antibodies, which have been correlated with improved patient outcomes[71–73]. To determine if the elicited

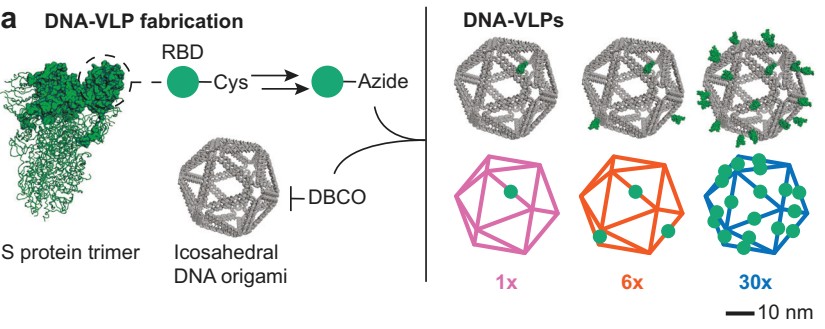
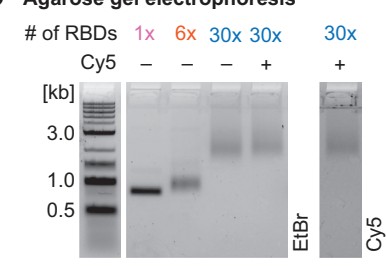
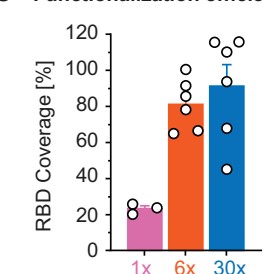
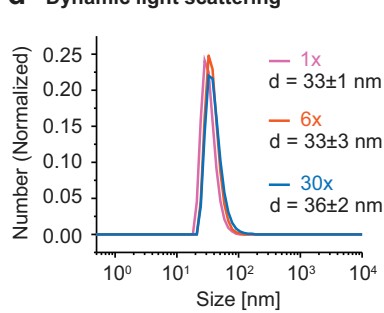
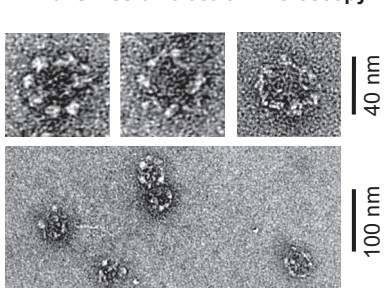

**Fig. 1 | Design and synthesis of DNA-VLPs covalently displaying the SARS-CoV-2 RBD. a** Recombinant RBD bearing an additional Cys residue at the C-terminus was expressed. The C-terminal Cys was selectively labeled with an SMCC-azide linker and subsequently conjugated to DNA-VLPs displaying DBCO groups. The icosahedral DNA origami objects of ~40 nm diameter displaying 1, 6, and 30 copies of the RBD were fabricated. **b** Agarose gel electrophoresis (AGE) shows the gel shift due to increasing RBD copy number as well as low polydispersity of the DNA-VLPs samples after purification. An additional VLP bearing 5 copies of Cy5 was produced for ACE2 binding flow cytometry experiments. Images are representative of n = 3 biological

replicates. **c** The coverage of the DNA-VLPs with RBD was quantified via Trp fluorescence. Coverage values were determined from n = 3 biological replicates for **DNA-VLP-1x** and from n = 6 biological replicates for DNA-VLP-6x and **DNA-VLP-30x**. **d** Dynamic light scattering (DLS) was used to assess the dispersity of functionalized VLP samples. Representative histograms are shown from n = 3 biological replicates. **e** Transmission electron micrographs (TEM) of **DNA-VLP-30x** were obtained by negative staining using 2% uranyl formate and validate the symmetric nanoscale organization of antigens. Images are representative of n = 3 technical replicates. Error bars and errors represent the standard error of the mean.

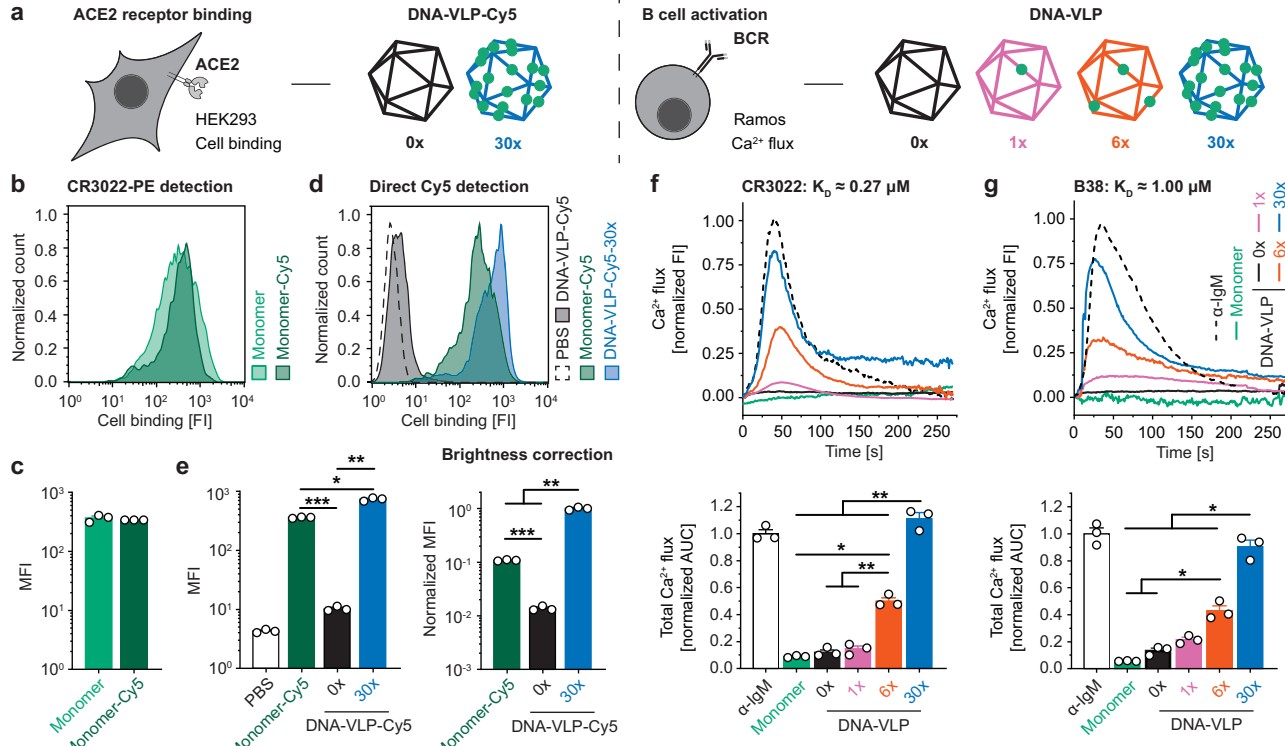

**Fig. 2 | In vitro activity of DNA-VLPs. a** An overview schematic of the in vitro activity assays and corresponding DNA-VLPs tested. **b, c** ACE2-expressing HEK 293 cells were incubated with 200 nM RBD. Binding was detected in flow cytometry experiments using PE-labeled CR3022 and a PE-labeled secondary antibody, demonstrating preserved binding activity for chemically modified monomeric RBD-Cy5 compared to monomeric RBD. Representative histograms are shown for ACE2 binding assays from n = 3 biological replicates, and median fluorescent intensity (MFI) values were determined from n = 3 biological replicates.

**d, e** Incubation with Cy5-labeled **DNA-VLP-Cy5-30x** at 100 nM RBD revealed enhanced binding compared to monomeric RBD-Cy5, likely due to multivalency effects. No unspecific binding for non-functionalized **DNA-VLP-Cy5** was observed. The brightness of Cy5-labeled **DNA-VLP-Cy5-30x** (5 Cy5 per 30 RBDs) and RBD-Cy5 (1 Cy5 per 1 RBD) were quantified experimentally and MFI values were corrected accordingly. Representative histograms are shown for ACE2 binding assays from n = 3 biological replicates, and MFI values were determined from n = 3 biological replicates. **f, g** Ramos B cells expressing the BCRs CR3022 and B38 were incubated with α-IgM, monomeric RBD, or DNA-VLPs at 30 nM RBD. $Ca^{2+}$ flux in response to RBD incubation was assayed using Fura Red. Representative fluorescence intensity

(FI) curves are shown from n = 3 biological replicates (top). Total $Ca^{2+}$ flux was quantified via the normalized area under the curve (AUC) (bottom). Normalized AUC values were determined from n = 3 biological replicates. Error bars represent the standard error of the mean. Two-sided Welch's *t*-test was performed at α = 0.05 for (**c**). One-way ANOVA was performed followed by Dunnett's T3 multiple comparison test at α = 0.05 for (**e**, **f**, and **g**). For (**e**), Left: *p* = 0.0006 for Monomer-Cy5:**DNA-VLP-Cy5-0x**, *p* = 0.0161 for Monomer-Cy5:**DNA-VLP-Cy5-30x**, *p* = 0.0042 for **DNA-VLP-Cy5-0x:DNA-VLP-Cy5-30x**; Right: *p* < 0.0001 for Monomer-Cy5:**DNA-VLP-Cy5-0x**, *p* = 0.0036 for Monomer-Cy5:**DNA-VLP-Cy5-30x**, *p* = 0.0029 for **DNA-VLP-Cy5-0x:DNA-VLP-Cy5-30x**. For **f**, *p* = 0.0170 for Monomer:**DNA-VLP-6x**, *p* = 0.0020 for **DNA-VLP-0x:DNA-VLP-6x**, *p* = 0.0038 for **DNA-VLP-1x:DNA-VLP-6x**, *p* = 0.0085 for Monomer:**DNA-VLP-30x**, *p* = 0.0016 for **DNA-VLP-0x:DNA-VLP-30x**, *p* = 0.0019 for **DNA-VLP-1x:DNA-VLP-30x**, *p* = 0.0080 for **DNA-VLP-6x:DNA-VLP-30x**. For (**g**), *p* = 0.0448 for Monomer:**DNA-VLP-6x**, *p* = 0.0368 for **DNA-VLP-0x:DNA-VLP-6x**, *p* = 0.0156 for Monomer:**DNA-VLP-30x**, *p* = 0.0213 for **DNA-VLP-0x:DNA-VLP-30x**, *p* = 0.0264 for **DNA-VLP-1x:DNA-VLP-30x**, *p* = 0.0131 for **DNA-VLP-6x:DNA-VLP-30x**. *$p$ < 0.05; **$p$ < 0.01; ***$p$ < 0.001.

antibodies were protective, we passively transferred immune sera (post-boost 2 timepoint) into K18-hACE2 transgenic mice and then challenged with a SARS-CoV-2 WA1/2020 virus containing the D614G substitution (Supplementary Fig. 7). In this challenge model, RBD-directed immune sera and/or RBD monoclonal antibodies provide humoral protection[39,74], and we recapitulated this protective activity with immune sera from **DNA-VLP-30x** and monomeric RBD, as measured by reductions in weight loss, and viral RNA within the nasal turbinates, nasal wash, and in the lung[39] (Supplementary Fig. 7).

To confirm that the **DNA-VLP-30x** boosted, RBD-specific IgG titers were achieved via the TD route, we compared the same sequential immunization regime in wild-type C57BL/6 versus *Tcra*[−/−] mice that lack functional TCRs[75] (Fig. 3a). **DNA-VLP-30x** elicited robust IgM responses in both genotypes, whereas RBD-specific IgG titers were not boosted in *Tcra*[−/−] mice (Fig. 3e). Hence, IgG boosting following sequential immunization with DNA-VLPs was due to TD, or T cell dependent, recall of B cell memory.

## Comparison of scaffold-directed antibody responses with P-VLPs versus DNA-VLPs

To compare our findings on DNA-VLPs with protein-based materials, we employed ferritin-based P-VLPs with 24-valent display of RBDs (**P-VLP-24x**) on a scaffold of ~12 nm scaffold diameter (Supplementary Fig. 1)[32,39]. Following the validation of efficient B cell activation in vitro (Supplementary Fig. 8), C57BL/6 mice were sequentially immunized intraperitoneally with monomeric RBD, **DNA-VLP-30x**, or **P-VLP-24x** at doses equivalent to 7.5 μg RBD (Fig. 4a). Although RBD-specific IgG titers were enhanced for both **DNA-VLP-30x** and **P-VLP-24x** (Fig. 4b, c), **P-VLP-24x** also exhibited boosted IgG titers against the ferritin protein scaffold itself (Fig. 4b). In contrast, we did not observe boosting of DNA-specific IgG titers following immunization with DNA-VLPs, indicating an absence of B cell memory of the DNA-based scaffold (Fig. 4c and Supplementary Fig. 9). Importantly, this absence of DNA-specific IgG responses was also observed when a higher DNA dose was administered (**DNA-VLP-6x**) and when monomeric RBD was administered (Supplementary Fig. 9).

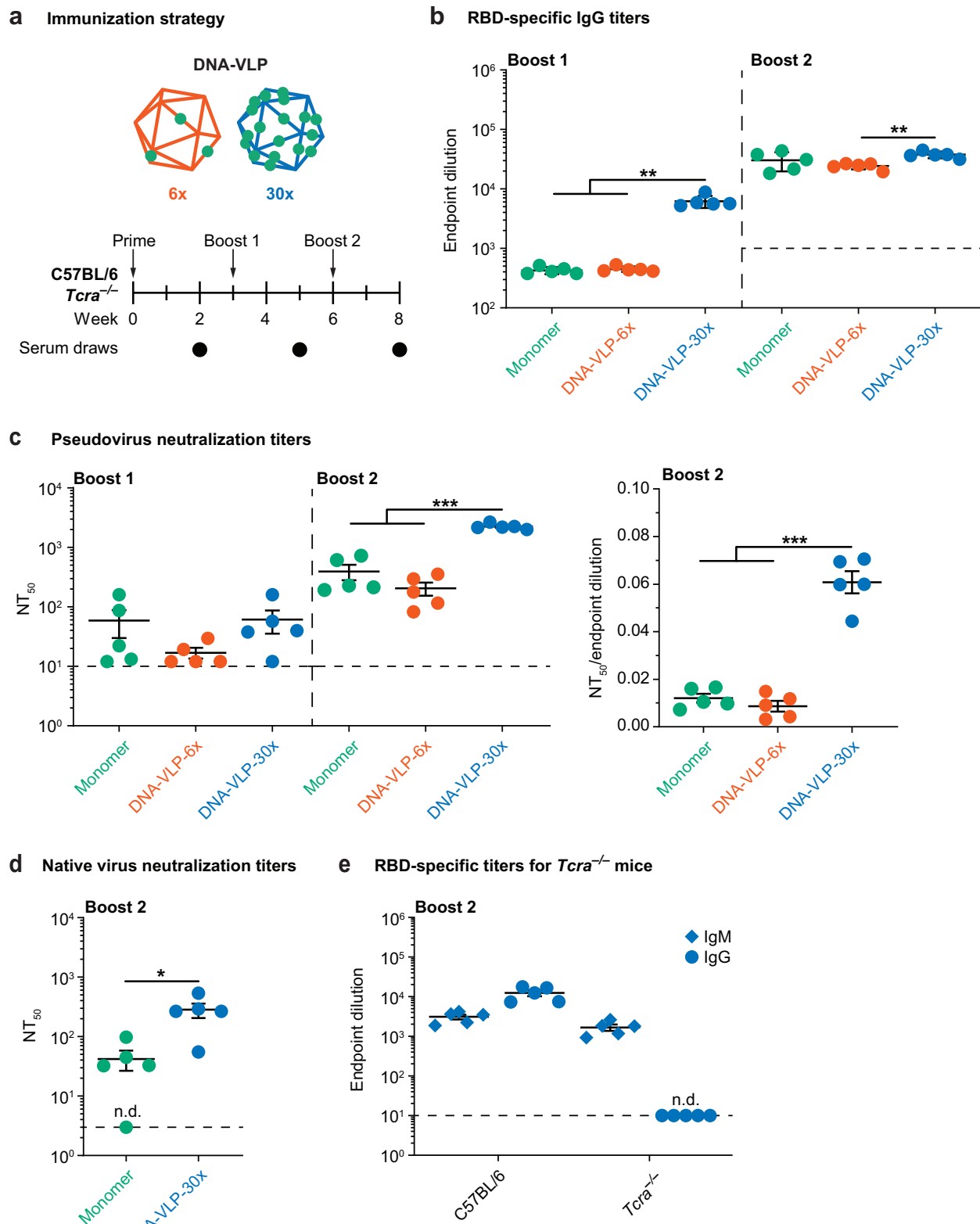

**Fig. 3 | RBD-specific antibody responses to DNA-VLPs. a** Mice were sequentially immunized with monomeric RBD and DNA-VLPs of varying copy number. **b** RBD-specific IgG endpoint dilutions revealed enhanced antibody responses for **DNA-VLP-30x** compared to both monomeric RBD and **DNA-VLP-6x**. **c** Serum neutralization titers expressed as $NT_{50}$ values against pseudoviruses modeling the ancestral Wuhan-1 strain. **d** Serum neutralization titers expressed as $NT_{50}$ against native Wuhan-CoV-2. **e** IgM and IgG titers of RBD-specific antibodies elicited in $Tcra^{-/-}$ and wild-type mice after sequential immunization with **DNA-VLP-30x**. N = 5 female mice were used in each experimental group. Error bars represent the standard error of the mean. Non-responder mice (denoted as n.d. = not detectable) were not considered for statistical analysis. One-way ANOVA was performed followed by Dunnett's T3 multiple comparison test at α = 0.05 for (**b**, **c**). Two-sided Welch's $t$-test was performed at α = 0.05 for (**d**). For (**b**), Boost 1: $p = 0.0021$ for Monomer:**DNA-VLP-30x**, $p = 0.0022$ for **DNA-VLP-6x:DNA-VLP-30x**; Boost 2: $p = 0.0048$ for **DNA-VLP-6x:DNA-VLP-30x**. For (**c**), Boost 2: $p = 0.0001$ for Monomer:**DNA-VLP-30x**, $p = 0.0001$ for **DNA-VLP-6x:DNA-VLP-30x**; Boost 2 Normalized: $p = 0.0005$ for Monomer:**DNA-VLP-30x**; $p = 0.0002$ for **DNA-VLP-6x:DNA-VLP-30x**. *$p < 0.05$; **$p < 0.01$; ***$p < 0.001$.

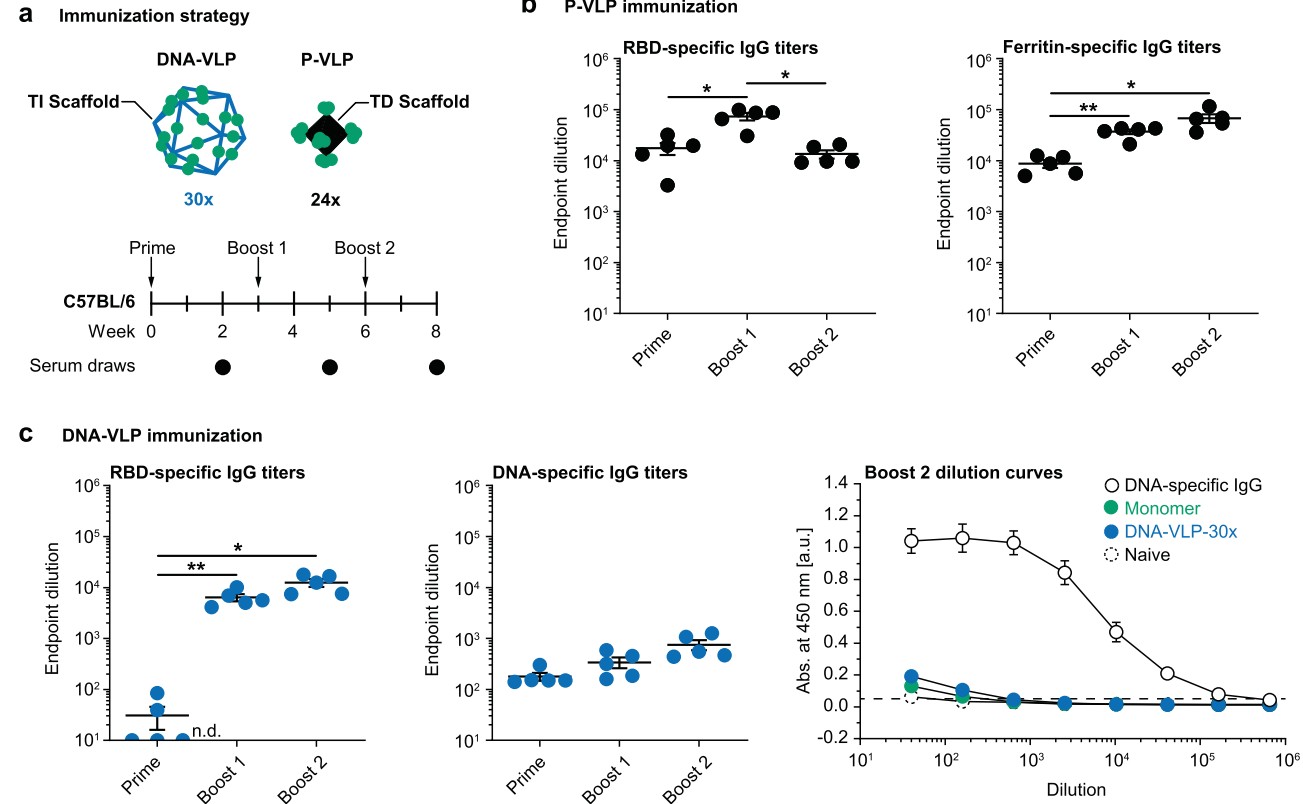

**Fig. 4 | Scaffold-specific antibody responses to DNA-VLPs and P-VLPs. a** Mice were sequentially immunized with monomeric RBD, **DNA-VLP-30x**, or ferritin-based **P-VLP-24x. b** RBD-specific and scaffold-specific IgG endpoint dilutions for the P-VLP immunization. **c** RBD-specific and scaffold-specific IgG endpoint dilutions for the DNA-VLP immunization and dilution curves for Boost 2 of the DNA-specific IgG ELISA. The DNA-specific IgG control was diluted from 10 μg/ml. N = 5 male mice were used in each experimental group. Error bars represent the standard error of the mean. Non-responder mice (denoted as n.d. = not detectable) were not considered for statistical analysis. One-way ANOVA was performed followed by Dunnett's T3 multiple comparison test at α = 0.05 for (**b**, **c**). For (**b**), RBD-specific IgG titers: $p = 0.0199$ for Prime:Boost 1, $p = 0.0209$ for Boost1:Boost 2; Ferritin-specific IgG titers: $p = 0.0036$ for Prime:Boost 1, $p = 0.0291$ for Prime:Boost 2. For (**c**), $p = 0.0094$ for Prime:Boost 1, $p = 0.0127$ for Prime:Boost 2. *$p < 0.05$; **$p < 0.01$; ***$p < 0.001$.

## Discussion

Rational vaccine design often seeks to leverage multivalent antigen display together with epitope-focusing to generate potent, neutralizing antibody responses[76]. However, the protein composition of P-VLPs results in the expansion of high titer antibody responses against both the displayed protein antigen and the protein scaffold itself[39,40,77]. Our findings suggest that a TI scaffold can mitigate this issue of anti-vector antibodies by ensuring: (1) valency-dependent antibody boosting against the scaffolded protein antigen within the TD pathway; and (2) confinement of scaffold-directed immunity to the TI pathway, which lacks a B cell memory response.

Our results suggest that DNA-VLPs covalently functionalized with 30 copies of the RBD antigen significantly enhance neutralizing antibody responses compared with monomeric RBD antigen alone, consistent with findings for P-VLPs displaying RBD[10,30,32,33,38,39], as well as other antigens[8,9,23]. However, unlike P-VLPs that elicit scaffold-directed humoral immunity within the memory compartment[32,33,38,39], as we observed here for a ferritin-based P-VLP, DNA-VLPs did not generate B cell memory to the VLP scaffold material itself. This suggests that the distraction of B cell memory away from the target protein antigen of interest in P-VLPs that potentially results in irreversible off-target immune imprinting[43,78–80] can be avoided by DNA-VLPs. While our finding was expected for TI antigens such as DNA, it is also well established that TD antibody responses can be generated for TI hapten antigens via covalent attachment to protein scaffolds[81,82]. Our results, however, indicate that scaffolding protein on TI antigens does not enable boostable B cell memory. At the same time, we observed robust valency-dependent TD antibody responses to the TD antigen, akin to virosomal and ISCOM-based vaccine designs, in which protein antigens are multivalently displayed by TI antigen-composed matrices[83–86].

Because DNA origami uniquely offers simultaneous yet independent control over spatial antigen display, scaffold size, and scaffold geometry, we were able to investigate the impact of antigen valency alone on antibody responses using an optimally sized, geometrically fixed ~34 nm icosahedral DNA-VLP scaffold, whereby 30 but not six copies of RBD were found to be sufficient to enhance neutralizing antibody titers compared with monomeric RBD alone. Future work may seek to examine the impact of antigen spacing with fixed valency[53], as well as alternative scaffold sizes and geometries, which might ultimately be required to resolve critical thresholds for enhancing antibody responses beyond monomeric antigens, as well as to optimize B cell responses for certain antigens[9,29,30]. Additional important extensions to our study include examining GC formation in B cell follicles, which are important for generating broadly neutralizing antibodies and long-lived humoral immunity[87–89]. Toward this end, it will be interesting to investigate to what extent multivalent antigen display by DNA-VLPs is maintained in secondary lymphoid organs, particularly in the presence of endonuclease[90,91] and protease[89] degradation; what the breadth of neutralizing antibody responses is; and what the longevity of humoral immunity is. To further enhance B cell activation and trafficking to secondary lymphoid organs, DNA-VLP valency may be increased and active follicle targeting with carbohydrates may be incorporated[5,87]. Co-formulation with TLR-based adjuvants could also enhance T cell help, drive GC reactions, and augment

humoral immunity[92–94]. Finally, DNA stabilization strategies[91] may be needed to increase the longevity of DNA-VLPs within follicles and GCs in the presence of endonucleases.

Beyond rational vaccine design and promoting antibody focusing, our discovery that DNA scaffolds are not neutralized by DNA-specific antibodies is of significant importance not only for vaccines, but also potentially for therapeutic nucleic acid delivery, as it enables redosing without antibody-dependent clearance[44,45].

## Methods

### Ethics statement
The research conducted complies with ethical regulations and biosafety approved by institutional committees at Mass General Brigham (MGB Institutional Biosafety Committee, protocols #2014B000035, 2018B000030; MGB Institutional Animal Care and Use Committee, protocol #2014N000252); MIT Institutional Biosafety Committee (protocol #BRR673); Washington University Institutional Biosafety Committee (protocol #14366); Washington University Animal Care and Use Committee (protocol #21-0246). This research does not include human participants.

### Materials
SS320 E. coli cells were purchased from Lucigen. EndoFree GigaPrep Kits (Cat. No. 12391) were purchased from Qiagen. Tris acetate-EDTA (TAE) and PBS buffer were purchased from Corning. Oligonucleotide staples (purified via desalting) were purchased from IDT. The standard base and dibenzocyclooctyne-triethylene glycol (DBCO-TEG) phosphoramidites as well as ancillary chemicals were purchased from Glen Research. Organic solvents were purchased from Sigma-Aldrich and VWR International. Agarose was purchased from IBI Scientific. The DNA agarose gel electrophoresis (AGE) standard (Quick-Load Purple 2-Log DNA ladder 0.1–10 kb) was purchased from New England Biolabs. $MgCl_2$, NaCl, $PEG_{8kDa}$, antibiotics, 2x YT medium, TE buffer, ethidium bromide (EtBr), TritonX-144, and 30% ammonium hydroxide (aqueous) was purchased from Sigma-Aldrich. Amicon Ultra centrifugal filters (10 kDa and 100 kDa) and dialysis membranes (mixed cellulose ester, 0.025 μm) were purchased from Sigma-Aldrich. Zeba spin columns (7 kDa) and Proteinase K were purchased from ThermoFisher Scientific. Transmission electron microscopy grids (CF200H-CU) were obtained from Electron Microscopy Sciences. Uranyl formate was purchased from Sigma-Aldrich. The codon-optimized gene for the expression of the SARS-CoV-2 receptor binding domain (RBD) was purchased form IDT. QuickChange Mutagenesis Kits (Cat. No. 200518) were purchased from Agilent. Expi293 Expression System Kits (Cat. No. A14635) were purchased from ThermoFisher Scientific. TALON cobalt resin was purchased from Takara. Nap-5 and Superdex 200 Increase columns were purchased from Cytiva. Ellman's reagent was purchased from Sigma-Aldrich. Amino-TEG-azide was purchased from BroadPharm. SMCC was purchased from ThermoFisher Scientific. ToxinSensor Gel Clot Endotoxin Assay Kits (Cat. No. L00351) were purchased from GenScript. 96- and 384-well cell culture plates were purchased from Corning and Greiner, respectively. 96-well MaxiSorp plates were purchased from Nunc. Sheep anti-human IgG Horseradish Peroxidase (HRP)-linked whole antibody and sheep anti-mouse IgG-HRP-linked whole antibody were purchased from Sigma-Aldrich. Goat F(ab')2 anti-human IgM-UNLB, goat anti-mouse IgM-HRP and mouse anti-human IgM-HRP were purchased from SouthernBiotech. Mouse anti-dsDNA antibody was purchased from Abcam. Mouse anti-human-kappa-light chain-PE was purchased from ThermoFisher. Mouse anti-human IgM APC, mouse anti-SARS-coV-2 Nucleocapsid and rat anti-mouse IgG2b-Pe-Cy7 were purchased from BioLegend. Calf-thymus DNA was purchased from Sigma-Aldrich. RPMI 1640 was purchased from Sigma-Aldrich. Cy5-Azide and Cy5-DBCO were purchased from Sigma-Aldrich. 1% casein blocking buffer was purchased from G-Biosciences. 100x penicillin-streptomycin-glutamine solution was

purchased from ThermoFisher Scientific. For neutralization assays, Vero-E6 (Cat. No. CRL-1586, ATCC) and A549-hAce2 (Cat. No. NR-53821, BEI Resources) cell lines were maintained in D10+ media (DMEM (Corning) supplemented with HEPES (Corning)), 1x Penicillin 100 IU/ml and Streptomycin 100 μg/ml (Corning), 1x Glutamine (Glutamax, ThermoFisher Scientific), and 10% fetal bovine serum (FBS) (Sigma-Aldrich) at 37 °C and 5% $CO_2$. SARS-CoV-2 USA-WA1/2020 viral stock was expanded from BEI Resources reagent NR-52281 on Vero-E6 cells and quantified by plaque assay. CR3022 anti-RBD IgG and B38 anti-RBD IgG antibodies were generated in-house[59,62].

### Scaffold synthesis
The custom-length DNA scaffold (Supplementary Table 1) for DNA-VLPs was prepared as previously performed[90]. Briefly, SS320 E coli cells were transformed with the phI52 plasmid, based on the pUC19 vector, and the M13cp helper plasmid (generously provided by Andrew Bradbury, Los Alamos National Laboratories). Next, pre-cultures of the transformed cells were grown overnight at 37 °C, diluted 100-fold and incubated for another 8 h, with all steps using 2x YT medium containing 100 μg/ml ampicillin, 15 μg/ml chloramphenicol and 5 μg/ml of tetracycline. Cells were sedimented by centrifuging three times at 4000 g for 3 min and subsequently discarded. Phage was precipitated from the supernatant in presence of 6% (w/v) $PEG_{8kDa}$ and 3% (w/v) of NaCl by stirring at 4 °C for 1 h and harvested by centrifugation at 20,000 g at 4 °C for 1 h. After resuspension in TE buffer, ssDNA was extracted via the EndoFree GigaPrep purification protocol with the following modifications: Proteinase K was added to buffer P1 followed by incubation at 37 °C for 1 h, addition of buffer P2 and incubation at 70 °C for 10 min. After ssDNA purification, Triton X-144 was used to remove residual endotoxins to levels less than 0.2 EU/μmol DNA scaffold, corresponding to less than 0.000015 EU/injection into mice[95]. Endotoxin levels were measured using ToxinSensor Gel Clot Endotoxin Assay Kits. Purity of the scaffold was analyzed by agarose gel electrophoresis (AGE) (1.6% agarose, TAE buffer with 12 mM $MgCl_2$, EtBr, 65 V for 150 min at 4 °C).

### Oligonucleotide staple synthesis
Solid-phase DNA synthesis was performed on a Dr. Oligo synthesizer (Biolytic). DNA synthesis was performed on a 200 nmol scale, starting from universal 1000 Å CPG solid supports and following the standard protocol[96]. Standard base and DBCO-TEG phosphoramidites were dissolved in anhydrous acetonitrile to afford 0.1 M solutions and were used in 10-fold excess. For the 5′ DBCO-TEG coupling, a coupling time of 10 min was used. Coupling efficiency was monitored after removal of the dimethoxy trityl (DMT) 5′-OH protecting groups. After solid-phase synthesis, oligonucleotides were cleaved off the resin in 30% ammonium hydroxide (aqueous) at 60 °C for 2 h, desalted with acetonitrile, and eluted with nuclease-free water. Installation of the DBCO-TEG modification was characterized by reversed-phase high-performance liquid (HPLC) and modified staples were purified as previously performed (BEH-$C_{18}$ column; 30 °C; 0.1 M triethylammonium acetate in water:acetonitrile gradient)[48]. For the assembly of **DNA-VLP-Cy5** and **DNA-VLP-Cy5-30x-DBCO**, Cy5-modified oligonucleotide staples were synthesized. 10x excess of Cy5-Azide was added to 50 μM DBCO-modified oligonucleotide staple in PBS with 10% DMF and incubated overnight at room temperature. Excess dye was removed using NAP-5 columns prior to purification of Cy5-modified oligonucleotide staples via reversed-phase HPLC (BEH-$C_{18}$ column; 30 °C; 0.1 M triethylammonium acetate in water:acetonitrile gradient).

### Antigen synthesis
The codon-optimized gene for the expression of the RBD of SARS-CoV-2 (GenBank ID = MN975262.1; residues 319-529) was cloned into a pVRC vector containing a C-terminal HRV C3 protease cleavage site followed by 8x His and SBP tags. An additional C-terminal Cys residue was

inserted using QuickChange Mutagenesis following the manufacturer's protocol and mutagenesis was confirmed by next-generation sequencing (Azenta) to afford RBD-Cys. HEK Expi293F cells were transiently transfected with the RBD plasmid using Expifectamine following the manufacturer's protocol. After 5–7 days, supernatants were harvested by centrifugation at 4000 g at room temperature for 5 min and the RBD-Cys was purified into PBS by affinity chromatography using TALON cobalt resin followed by size-exclusion chromatography using Superdex 200 Increase columns and stored at 4 °C for less than 7 days.

Antigen modification with the azide linker was adapted from published protocols[48]. Briefly, 100 μM RBD-Cys was incubated with 10x excess of TCEP in PBS for 30 min at room temperature and subsequently purified into PBS with 10 mM EDTA using Zeba spin columns (7 kDa). Incubation for ~6 h at room temperature allowed for the reoxidation of disulfides prior to the addition of the azide linker as monitored by Ellman's assay. The azide linker was assembled by incubation of 60 mM SMCC with 1.1x excess of amino-TEG-azide for 1 h at room temperature. Subsequently, 10x excess of crude azide linker was added to reduced antigen and the reaction mixture was incubated overnight at room temperature to afford RBD-Az. RBD-Az was purified into PBS using Amicon Ultra centrifugal filters (10 kDa, 5000 g) followed by size-exclusion chromatography using Superdex 200 Increase columns and stored at 4 °C for less than 7 days. To label azide-modified antigen with dyes, 50 μM RDB-Az was incubated with 5x excess of DBCO-Cy5 in PBS for 30 min at room temperature and subsequently purified into PBS using Amicon Ultra centrifugal filters (10 kDa, 5000 g). RBD concentrations and dye labeling efficiency were determined by absorbance measurements at 280 nm ($\varepsilon = 39400$ 1/(M•cm) and MW = 31 kDa) and 650 nm ($\varepsilon = 250000$ 1/(M•cm)).

## DNA-VLP design and assembly
**DNA-VLP-0x-DBCO, DNA-VLP-1x-DBCO, DNA-VLP–6x-DBCO**, and **DNA-VLP–30x-DBCO** as well as **DNA-VLP-5x-Cy5** and **DNA-VLP-5x-Cy5-30x-DBCO** were designed using DAEDALUS and nick position of edge staples were adjusted manually for outward orientation (Supplementary Tables S2, 3)[47]. The DNA-VLPs were assembled as previously performed[47]. Briefly, 30 nM of scaffold and 300 nM of each oligonucleotide staple were dissolved in TAE buffer with 12 mM MgCl₂ and thermally annealed as follows: 95 °C for 5 min, 80–75 °C at 1 °C per 5 min, 75–30 °C at 1 °C per 15 min, and 30–25 °C at 1 °C per 10 min. The DNA-VLPs were purified into PBS using Amicon Ultra centrifugal filters (100 kDa, 2000 g) and stored at 4 °C. Purity and monodispersity of the DNA-VLPs were validated by AGE (1.6% agarose, TAE buffer with 12 mM MgCl₂, EtBr, 65 V for 150 min at 4 °C) (Supplementary Fig. 1).

## DNA-VLP functionalization
DBCO-bearing DNA-VLPs were functionalized with RBD-Az to yield **DNA-VLP-1x, DNA-VLP-6x, DNA-VLP-30x** or **DNA-VLP-Cy5-30x**. At least 150 nM of **DNA-VLP-30x-DBCO** or **DNA-VLP-Cy5-30x-DBCO** and at least 750 nM of **DNA-VLP-1x-DBCO** or **DNA-VLP-6x-DBCO** were incubated with 30 equivalents per DBCO group of RBD-Az in PBS at room temperature for 24 h. RBD-functionalized DNA-VLPs were purified into PBS by drop dialysis (mixed cellulose ester, 0.025 μm) and stored at room temperature for less than 7 days. Purity and monodispersity of the DNA-VLPs were validated by AGE (1.6% agarose, TAE buffer with 12 mM MgCl₂, EtBr, 65 V for 150 min at 4 °C) and dynamic light scattering (DLS, Malvern Zetasizer Ultra). Coverage with antigens was measured using Trp fluorescence spectroscopy[53] at 100 nM RBD and denaturing, reversed-phase HPLC as previously performed (BEH-C₁₈ column; 30 °C; 0.1 M triethylammonium acetate in water: acetonitrile gradient)[48].

## Transmission electron microscopy
Uranyl formate staining of DNA-VLP samples was adapted from an existing protocol[97]. Briefly, **DNA-VLP-30x** was diluted to 5 nM, and 5 μl

of the solution were immediately deposited onto glow-discharged electron microscopy grids. After 30 s, the solution was removed by blotting with filter paper and the grids were washed with 5 μl of freshly prepared 2% uranyl formate with 5 mM NaOH. After removal of the washing solution by blotting, 15 μl of the uranyl formate solution was added, incubated for 30 s and the removed by blotting. Finally, the grids were dried in vacuo and transmission electron microscopy (TEM, FEI Tecnai G2 Spirit Twin) was conducted at 120 keV.

## Protein-based virus-like particle synthesis
The SARS-CoV-2 RBD with C-terminal 8x His and SpyTags was expressed and purified as described above. *H. pylori* ferritin nanoparticles were expressed with N-terminal SpyCatcher and 8x His tags were expressed and purified as previously performed[39]. SpyTag-SpyCatcher conjugation was performed overnight at 4 °C at 4x excess of RBD-SpyTag per SpyCatcher. 24-valent P-VLPs bearing RBD (**P-VLP-24x**) were subsequently purified into PBS by size-exclusion chromatography to remove excess RBD SpyTag as previously performed[39].

## Biolayer interferometry
Biolayer interferometry binding experiments were performed after immobilization of CR3022 and B38 Fab at 0.1 mg/ml on FAB2G sensors (BLItz, Sartorius). Wild-type RBD and RBD-Az served as analytes at 10 μM in the manufacturer's Kinetics Buffer.

## ACE2-expressing cell binding assay
ACE2 expressing HEK 293T cells (generously provided by Nir Hacohen and Michael Farzan, Massachusetts General Hospital and The Scripps Research Institute) were harvested and washed with PBS with 2% FBS. 200,000 cells per well were transferred to 96-well cell culture plates and 100 μl of wild-type RBD and RBD-Cy5 at concentrations corresponding to 200 nM RBD in PBS were added. Following incubation for 60 min on ice, cells were washed twice with PBS with 2% FBS and stained with 50 μl at 200 nM of the anti-RBD antibody CR3022 for 30 min at room temperature, following pre-complexation with goat anti-human-PE at 200x excess. The suspension was protected from light and incubated for 30 min on ice, washed twice with PBS with 2% FBS and resuspended in 100 μl PBS with 2% FBS. For RBD-Cy5, **DNA-VLP-Cy5**, and **DNA-VLP-Cy5-30x**, cell binding was also detected via direct Cy5 fluorescence. After incubation with RBD-Cy5 or DNA-VLPs at concentrations corresponding to 100 nM RBD (**DNA-VLP-Cy5** concentration was equivalent to **DNA-VLP-Cy5-30x**), no staining was performed and cells were directly resuspended in 100 μl PBS with 2% FBS. Cell binding was analyzed by flow cytometry (S1000Exi Flow Cytometer, Strategim) and data processing was conducted using FlowJo (BD Biosciences, v.10). Cy5 fluorescence intensities obtained by flow cytometry were corrected according the relative brightness of RBD-Cy5, **DNA-VLP-Cy5**, and **DNA-VLP-Cy5-30x** as quantified by fluorescence spectroscopy in PBS.

## B cell activation assay
The B cell activation assay was adapted from previously established protocols[61,69,98]. Briefly, the human anti-RBD antibodies CR3022[59] and B38[62] were expressed as IgM B cell receptors (BCRs) in Ramos B cells after lentiviral transfection of the corresponding light chain and transmembrane IgM heavy chain genes. 5 to 7 days after transfection, BCR-expressing Ramos B cells were FACS-sorted for IgM and κ light chain expression (SH800S Cell Sorter, Sony Biotech). Sorted cells were expanded in RPMI supplemented with 15% FBS and 1x penicillin-streptomycin-glutamine and 1,000,000 cells were harvested and resuspended RPMI with Fura Red solution following the manufacturer's protocol. After incubation for 20 min at 37 °C, cells were harvested and resuspended in 500 μl RPMI medium prior to detection of BCR signaling by flow cytometry at 637 nm. Following 30 s of

baseline data acquisition, wild-type RBD, **DNA-VLP-0x, DNA-VLP-1x, DNA-VLP-6x, DNA-VLP-30x**, or **P-VLP-24x** at concentrations corresponding to 30 nM RBD were added before continuing data acquisition for an additional 270 s. The concentration of **DNA-VLP-0x** corresponded to that of **DNA-VLP-1x**. Goat anti-human IgM at 10 µg/ml served as a positive control. Maximum total $Ca^{2+}$ flux was measured after addition of 10 µg/ml ionomycin. Fluorescence traces were processed as follows: For each trace, the average fluorescence of the 30 s baseline data acquisition was subtracted. Next, the fluorescence traces were normalized to the $Ca^{2+}$ flux induced by ionomycin to obtain relative $Ca^{2+}$ flux traces. The total $Ca^{2+}$ flux was quantified by integration to obtain the normalized area under the curve (AUC).

## Immunization experiments

Wild-type C57BL/6 or $Tcra^{-/-}$ mice[75] (n = 5 per group, males and females, 6–8 weeks of age) were pre-bled and then sequentially immunized intraperitoneally with wild-type RBD, **DNA-VLP-6x, DNA-VLP-30x**, or **P-VLP-24x** at equimolar doses equivalent to 7.5 µg RBD. The antigens were injected in 100 µl containing 50% Sigma adjuvant as performed previously[34,64,98]. Immunization occurred at weeks 0, 3, and 6 and blood draws occurred 2 weeks after each immunization. The animals were maintained within the Ragon Institute's HPPF barrier facility and the experiments were conducted with IACUC approval (MGH protocol 2014N000252). The light cycles in the animal room were set on a 12 h light cycle [7AM-7PM (ON) 7PM-7AM (OFF)]. The temperature range for the room was 68–73 °F and the humidity index was from 30–70%. At the end of experiments, the animals were euthanized by $CO_2$ inhalation (30% of the chamber volume/min).

## RBD- and ferritin-specific IgG ELISA

Recombinant RBD (or ferritin) was used to coat MaxiSorp plates at 200 ng/well overnight at 4 °C. The plates were washed with PBS, blocked with PBS with 3% non-fat milk for 1 h at room temperature and subsequently washed with PBS. Mouse sera or human anti-RBD antibodies were diluted in PBS, transferred to the plates and incubated for 1 h at room temperature. The plates were washed with PBS with 0.05% Tween-20 and subsequently incubated with a 1:5000 dilution of either goat anti-mouse-IgM (IgG-HRP, GE Healthcare) or sheep anti-murine IgG (HRP-IgG, GE Healthcare). Following washing with PBS with 0.05% Tween-20, 3,3′,5,5′ tetramethylbenzidine (TMB) was added, and the developer reaction was then stopped using 1 N sulfuric acid. The plates were read by absorbance at 450 nm (Infinite m1000 Pro microplate absorbance reader, Tecan). Antibody endpoint dilutions were calculated using an absorbance cut-off of 0.05 (Graphpad Prism v.9.1, GraphPad Software Inc). The loading of RBD was also standardized to recombinant mAbs CR3022 and B38, as expressed and purified previously[39,73,99]. In this case ELISA was performed as above, except that sheep anti-human IgG-HRP (GE Healthcare) was used as the secondary antibody.

## DNA-specific IgG ELISA

Calf-thymus DNA was reconstituted in water 50 µg/ml and used to coat 96-well MaxiSorp plates overnight at 4 °C. The plates were washed with PBS, blocked with 1% casein buffer for 2 h at room temperature and subsequently washed with PBS. Mouse sera (1:30) or mouse anti-dsDNA antibody (1:100) were diluted in blocking buffer, transferred to the plates and incubated for 2 h at room temperature. The plates were washed with PBS with 0.2% Tween-20 and incubated with 1:5000 dilution goat anti-mouse-IgG-HRP antibody (BioRad) for 1 h at room temperature. Following washing with PBS with 0.2% Tween-20, TMB was added, the plates were incubated for 3 min and the reaction was stopped using sulfuric acid. DNA-specific IgG titers were determined by absorbance measurements at 450 nm. Endpoint dilutions were calculated using an absorbance cut-off of 0.05 as determined from the limit of detection determined

for PBS-incubated wells (Graphpad Prism v.9.1, GraphPad Software Inc).

## Pseudovirus neutralization assay

SARS-CoV-2 neutralization was assessed using pseudotyped lentivirus particles expressing S glycoprotein trimer as previously described[71]. Briefly, pseudovirus corresponding to the ancenstral Wuhan-1 strain were produced by transient transfection of HEK 293T cells. The titers of viral supernatants were determined via flow cytometry with ACE2-expressing HEK 293T cells and via the HIV-1 p24$^{CA}$ antigen capture assay (Leidos Biomedical Research). Assays were performed in 384-well plates using a fluorescence plate reader (Tecan Fluent Automated Workstation). Mouse sera (or CR3022 and B38 mAb standards), starting from a 3x initial dilution, were serially 3x diluted in 20 µl followed by the addition of 20 µl of pseudovirus containing 250 IFU and incubated for 1 h at room temperature. Next 10,000 ACE2-expressing HEK 293T cells are were added per well and incubated for 60 to 72 h at 37 °C. After transfection, cells were lysed and incubated on a shaker for 5 min at room temperature before measuring luciferase expression (Molecular Devices SpectraMax L). Relative neutralization for each serum dilution was calculated after subtracting background luminescence and dividing by the luminescence in absence of sera. $NT_{50}$ vales were derived by fitting the Dose-Response equations to serum dilution curves.

## Authentic virus neutralization assay

SARS-CoV-2 neutralization was also assessed using authentic virus as previously described[73]. Briefly, A549-hACE2 cells were detached using Trypsin-EDTA (ThermoFisher Scientific) and seeded at 40,000 cells per well in 96-well plates 16–20 h before infection. Four hours before infection, the cell culture supernatant was removed and 75 µL of D2+ media was added (2% FBS instead of 10%).

Mouse sera were diluted in D+ media (no FBS) in 3-fold serial dilutions, mixed 1:1 (v/v) with SARS-CoV-2 diluted at 40,000 pfu/ml and incubated at 37 °C and 5% $CO_2$ for 1 h. 25 µl of the sera-virus solutions were added in triplicate wells for each condition for a final multiplicity of infection of 0.01 (virus to cell ratio). The 96-well plates were centrifuged for 30 min at 2000 g at 37 °C and then incubated at 37 °C and 5% $CO_2$ for 48 h. Each plate included a no infection control, a no treatment control for maximum infection, as well serially diluted B38 mAb as a positive control.

After 48 h, the cell culture supernatant was discarded, the cells were washed with PBS (Corning) then harvested using TrypLE (Life Technologies) and flow cytometry buffer (2% FBS in PBS). Cells were washed, then stained with live/dead fixable blue stain (ThermoFisher Scientific) for 30 min at 4 °C. After one wash with flow cytometry buffer, the cells were fixed using 4% paraformaldehyde (Santa Cruz) for 30 min at 4 °C. The fixed cells were removed from the BSL3 laboratory and prepared for intracellular staining using Perm/Wash buffer (BD Biosciences). The permeabilized cells were stained with mouse anti-SARS-CoV-2 Nucleocapsid antibody (Biolegend) for 30 min at 4 °C, then washed and stained with secondary antibody labeled with Pe-Cy7 (Biolegend) for 30 min at 4 °C. Finally, the cells were washed and resuspended in flow cytometry buffer.

Flow cytometry was performed on a BD Symphony (BD Biosciences). FCS files were analyzed using FlowJo software (v.10, BD Biosciences). Additional data analysis was performed using GraphPad Prism (v.9.1, GraphPad Software Inc) to fit curves for the neutralization data.

## Passive transfer and SARS-CoV-2 challenge

Challenge studies were carried out in accordance with the recommendations in the Guide for the Care and Use of Laboratory Animals of the National Institutes of Health. The protocols were approved by the Institutional Animal Care and Use Committee at the Washington

University School of Medicine (protocol #21-0246). Virus inoculations were performed under anesthesia that was induced and maintained with ketamine hydrochloride and xylazine, and all efforts were made to minimize animal suffering. Eight-week-old female heterozygous K18-hACE2 C57BL/6 J mice (strain: 2B6.Cg-Tg(K18-ACE2) 2Prlmn/J) were obtained from The Jackson Laboratory. All animals were housed in groups of 3 and fed standard chow diets. The photoperiod was 12 h on:12 h off dark/light cycle. The ambient animal room temperature was 70 °F, controlled within ±2° and the room humidity was 50%, controlled within ±5%. In vivo studies were not blinded, and mice were randomly assigned to treatment groups. No sample-size calculations were performed to power each study. Instead, sample sizes were determined based on prior in vivo virus challenge experiments. Mice were injected with 150 μl of naive or immune sera (from each immunized donor, post-boost 2 timepoint) via the intraperitoneal route. One day later, animals were intranasally inoculated with $10^3$ FFU of a WA1/2020 SARS-CoV-2 strain containing the D614G mutation[39,100]. Body weights were monitored daily. On day 6 post-challenge, the mice were sacrificed, and viral RNA was measured in the nasal turbinates, nasal wash fluid, and the lung[39].

To measure the viral RNA levels, tissues were weighed and homogenized with zirconia beads in a MagNA Lyser instrument (Roche Life Science) in 1 ml of DMEM medium supplemented with 2% heat-inactivated FBS. Tissue homogenates were clarified by centrifugation at -10,000 g for 5 min and stored at −80 °C. RNA was extracted using the MagMax mirVana Total RNA isolation kit (ThermoFisher Scientific, Cat. No. A27828) on the Kingfisher Flex extraction robot (Thermo-Fisher Scientific). RNA was reverse transcribed and amplified using the TaqMan RNA-to-CT 1-Step Kit (ThermoFisher Scientific, Cat. No. 4392653). Reverse transcription was carried out at 48 °C for 15 min followed by 2 min at 95 °C. Amplification was accomplished over 50 cycles as follows: 95 °C for 15 s and 60 °C for 1 min. Copies of SARS-CoV-2 N gene RNA in samples were determined using a previously published assay[101]. Briefly, a TaqMan assay was designed to target a highly conserved region of the N gene:

Forward primer: ATGCTGCAATCGTGCTACAA
Reverse primer: GACTGCCGCCTCTGCTC
Probe: /56-FAM/TCAAGGAAC/ZEN/AACATTGCCAA/3IABkFQ/

This region was included in an RNA standard to allow for copy number determination down to 10 copies per reaction. The reaction mixture contained final concentrations of primers and probe of 500 and 100 nM, respectively.

## Reporting summary

Further information on research design is available in the Nature Portfolio Reporting Summary linked to this article.

## Data availability

The source data used to generate Figs. 1–4 and Supplementary Figs. 1–9 are provided in the Source Data file. All other data are available in the article and its Supplementary files or from the corresponding author upon request. Source data are provided with this paper.

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

## Acknowledgements

E.-C.W., G.A.K., A.R. and M.B. were supported by NIH R21-EB026008, NSF CCF-1564025, CBET-1729397, and CCF-1956054, ONR N00014-21-1-4013 and N00014-17-1-2609, ARO ISN W911NF-13-D-0001, and Fast Grants Agreement EFF 4/15/20. E.-C.W. was additionally supported by the Feodor Lynen Fellowship of the Alexander von Humboldt Foundation. G.A.K. was additionally supported by the National Science Foundation under a Graduate Research Fellowship 4000189657. A.R. was additionally supported by the National Science Foundation under a Graduate Research Fellowship 4000168384. E.-C.W., G.A.K., A.R. and M.B. acknowledge MIT.nano for the use of core characterization facilities, and acknowledge support from core center grants P30-ES002109, NIEHS and P30-CA14051, NCI. B.M.H. was supported by NIGMS T32GM007753 and F30 AI160908. J.F. was supported by NIGMS T32AI007245. A.G.S. was supported by NIH R01 AI146779 and a Massachusetts Consortium on Pathogenesis Readiness (MassCPR) grant. D.L. was supported by NIH R01AI124378, R01AI153098, R01AI155447, U19AI057229, and P30AI060354. A.B.B. was supported by NIH R01AI174875, R01AI174276, DP2DA040254 and CDC subcontract 200-2016-91773-T.O.2 and a Massachusetts Consortium on Pathogenesis Readiness (MassCPR) grant. M.S.D. was supported by NIH R01AI157155. SARS-CoV-2 virus work was performed in part at the Ragon Institute's Biosafety Level 3 facility, which is supported by the NIH-funded Harvard University Center for AIDS Research (P30 AI060354) and the Massachusetts Consortium on Pathogen Readiness (MassCPR). The authors are also grateful to Vintus Okonkwo and Faez Amokrane Nait Mohamed in the Lingwood lab for technical assistance and maintenance of B cell lines and Benjamin Clancy for technical assistance on DNA origami materials production.

## Author contributions

Conceptualization: M.B., D.L., A.G.S. Methodology: E.-C.W., L.R., J.F., G.A.K., B.M.H., A.R., J.B.C., S.S., E.L., K.S.D., J.B. Investigation: E.-C.W., L.R., J.F., G.A.K., B.M.H., A.R., J.B.C. Visualization: E.-C.W., L.R., G.A.K. Supervision: M.B., D.L., A.G.S., M.S.D., A.B.B., A.K.B. Writing—original draft: E.-C.W., L.R., D.L., M.B. Writing—review & editing: E.-C.W., L.R. G.A.K., A.R., J.B.C., M.S.D., D.L., M.B.

## Competing interests

The Massachusetts Institute of Technology has filed a patent (US application number 16/752,394) covering the use of DNA origami as a vaccine platform on behalf of the co-inventors (E.-C.W. and M.B.). M.S.D. is a consultant for Inbios, Vir Biotechnology, Moderna, Merck, IntegerBio, and Immunome. The Diamond laboratory has received unrelated funding support in sponsored research agreements from Emergent BioSolutions, Moderna, Vir Biotechnology, and IntegerBio. D.L. reports SAB membership for Flagship Labs 72, Tendel Therapies, and Lattice Therapeutics Inc. M.B. is co-founder and SAB member of Kano Therapeutics, Inc. and Cache DNA, Inc. The remaining authors declare no competing interests.
