## [Peer Review File · Nature Communications]

Enhancing antibody responses by multivalent antigen display on thymus-independent DNA origami scaffoldsEditorial Note: This manuscript has been previously reviewed at another journal that is not operating a transparent peer review scheme. This document only contains reviewer comments and rebuttal letters for versions considered at *Nature Communications*.

REVIEWER COMMENTS

Reviewer #1 (Remarks to the Author):

The study reports the efficacy of DNA-VLPs for its multivalent antigen presentation and negligible off-target effects. The DNA-VLPs decorated with SARS-CoV-2 Spike protein elicited a T-cell-dependent humoral response. The sera collected from immunized animals efficiently neutralized different variants of SARS-CoV-2. In summary, the authors have conducted in vivo experiments to show the vaccine efficacy of DNA-VLPs decorated with SARS-CoV-2 Spike protein. The current study is an extension of a previous report (published in *Nature Nanotechnology* in 2020) that is of specific interest to coronavirus research but lacks the needed depth for publication in *Nature Communications*. The authors need to address the following concerns before this study can be recommended for publication.

Major Comments:

1. The study shows efficient pseudo-virus and authentic virus neutralization with antibody sera drawn from the mice. However, the actual protection remains unexplored. The authors need to perform prophylactic/therapeutic studies to demonstrate the in vivo efficacy of the DNA-VLP-RBS vaccine.
2. The authors also need to demonstrate any cross-reactivity of the mice-isolated antibody sera against other viruses, preferably influenza to demonstrate the specificity of the DNA-VLP-RBS vaccine.
3. The study lacks a systemic toxicity assay to determine the safety of the DNA-VLP-RBD vaccine in mice.

Minor Comments:

1. The authors need to clearly mention in the abstract that the icosahedron VLPs carry the SARS-CoV-2 spike protein RBD.
2. The authors report a T-cell-dependent antibody boosting. While not critical to the scope of the study, the author can report if the DNA-VLP-RBD vaccine also improves any T-cell responses against SARS-CoV-2, cytotoxic T-cell responses.

Reviewer #2 (Remarks to the Author):

445644_0

This paper has been revised and resubmitted. The text has been clarified and some new data has been added, mainly the evidence that the immune response to RBD protein presented in a multimeric form on a DNA origami scaffold is greatly reduced in TCR α $-/-$ mice. Overall, the study shows that this multimerization approach avoids excessive antibody production directed to the multimerization moiety. A control construct with RBD multimerized on Ferritin elicited significant amounts of antibodies to the Ferritin scaffold, demonstrating that anti-scaffold responses to protein could be a problem.

Given that the current COVID vaccines involve mRNA delivery of minimally modified spike protein that lacks a heterologous multimerization scaffold, the approach in this paper does not really address a need, nor is it likely it would be adopted as a COVID vaccine. This materially diminishes the impact of the paper. However, there may be an argument that the vaccine approach described here would have value for vaccinations to other protein subunits of interest.

Overall, the paper is acceptable with a modest degree of enthusiasm.

Reviewer #3 (Remarks to the Author):

The study reports the efficacy of DNA-VLPs for its multivalent antigen presentation and negligible off-target effects. The DNA-VLPs decorated with SARS-CoV-2 Spike protein elicited a T-cell-dependent humoral response. The sera collected from immunized animals efficiently neutralized different variants of SARS-CoV-2. In summary, the authors have conducted in vivo experiments to show the vaccine efficacy of DNA-VLPs decorated with SARS-CoV-2 Spike protein. The current study is an extension of a previous report (published in Nature Nanotechnology in 2020) that is of specific interest to coronavirus research but lacks the needed depth for publication in Nature Communications. The authors need to address the following concerns before this study can be recommended for publication.

Major Comments:

1. The study shows efficient pseudo-virus and authentic virus neutralization with antibody sera drawn from the mice. However, the actual protection remains unexplored. The authors need to perform prophylactic/therapeutic studies to demonstrate the in vivo efficacy of the DNA-VLP-RBS vaccine.

2. The authors also need to demonstrate any cross-reactivity of the mice-isolated antibody sera against other viruses, preferably influenza to demonstrate the specificity of the DNA-VLP-RBS vaccine.
3. The study lacks a systemic toxicity assay to determine the safety of the DNA-VLP-RBD vaccine in mice.

Minor Comments:

1. The authors need to clearly mention in the abstract that the icosahedron VLPs carry the SARS-CoV-2 spike protein RBD.
2. The authors report a T-cell-dependent antibody boosting. While not critical to the scope of the study, the author can report if the DNA-VLP-RBD vaccine also improves any T-cell responses against SARS-CoV-2, cytotoxic T-cell responses.

Reviewer #1 (Remarks to the Author in quotes and black font; Responses to the Reviewer in blue font; Revised manuscript text in red font):

Reviewer #1 (Remarks to the Author):

“The study reports the efficacy of DNA-VLPs for its multivalent antigen presentation and negligible off-target effects. The DNA-VLPs decorated with SARS-CoV-2 Spike protein elicited a T-cell-dependent humoral response. The sera collected from immunized animals efficiently neutralized different variants of SARS-CoV-2. In summary, the authors have conducted in vivo experiments to show the vaccine efficacy of DNA-VLPs decorated with SARS-CoV-2 Spike protein. The current study is an extension of a previous report (published in Nature Nanotechnology in 2020) that is of specific interest to coronavirus research but lacks the needed depth for publication in Nature Communications. The authors need to address the following concerns before this study can be recommended for publication.”

We thank the Reviewer for their summary of our work. Here, we would like to further highlight two impactful aspects of our work that are not mentioned in this summary. First, we observed that scaffolding thymus-dependent antigens (RBD) on a thymus-independent scaffold (DNA origami) led to antibody production against the on-target antigen, while mitigating off-target scaffold-directed antibodies; this is in contrast with data in our work that show protein-based nanoparticle subunit vaccines produce large off-target scaffold responses. We believe this discovery has important implications for vaccine design and delivery, as discussed in the Abstract and Discussion sections of our manuscript. Second, our study contributes to an early, but growing, literature of in vivo animal studies showing the utility of DNA origami-based materials for biomedical applications, with significant potential for medicine.

“Major Comments:

1. The study shows efficient pseudo-virus and authentic virus neutralization with antibody sera drawn from the mice. However, the actual protection remains unexplored. The authors need to perform prophylactic/therapeutic studies to demonstrate the in vivo efficacy of the DNA-VLP-RBS vaccine.”

We appreciate the Reviewer’s suggestion of exploring the protection capabilities of our materials. Towards this, we conducted a passive transfer and challenge experiment led by new senior author Dr. Michael S. Diamond (Washington University Division of Infectious Diseases), in which transgenic mice expressing the human ACE2 receptor were infused with immunized serum from the post-boost 2 timepoint, and then challenged with WA1/2020 SARS-CoV-2 containing the D614G mutation, as previously conducted^{1,2}. In this challenge model, RBD-directed immune sera and/or RBD monoclonal antibodies provide protection from the viral challenge. Our results demonstrate that DNA-VLPs do generate protective antibody responses in this model, as assessed by body weight, and by viral RNA levels in nasal turbinates and washes, and in the lung. These data have been incorporated into the Supplementary Information as **Figure S7**, and the corresponding description and discussion can be found in the Main Text and Methods section of our revised manuscript.

Edited (Page 2, line 37)

To determine if the elicited antibodies were also protective, we passively transferred immune sera (post-boost 2 timepoint) into K18-hACE2 transgenic mice and then

challenged with a SARS-CoV-2 virus containing the D614G mutation (**Figure S7**). In this challenge model, RBD-directed immune sera and/or RBD monoclonal antibodies provide humoral protection^{39,78}, and we recapitulate this protective activity with DNA-VLP-30x immune sera, as measured by weight loss, and viral RNA within the nasal turbinates, nasal wash, and in the lung³⁹ (**Figure S7**).

Edited (Page 17, line 522)

Passive transfer and SARS-CoV-2 challenge.

Passive transfer and SARS-CoV-2 challenge.

Challenge studies were carried out in accordance with the recommendations in the Guide for the Care and Use of Laboratory Animals of the National Institutes of Health. The protocols were approved by the Institutional Animal Care and Use Committee at the Washington University School of Medicine (protocol #21-0246). Virus inoculations were performed under anesthesia that was induced and maintained with ketamine hydrochloride and xylazine, and all efforts were made to minimize animal suffering. Eight-week-old female heterozygous K18-hACE2 C57BL/6 J mice (strain: 2B6.Cg-Tg(K18-ACE2) 2PrImn/J) were obtained from The Jackson Laboratory. All animals were housed in groups of 3 and fed standard chow diets. The photoperiod was 12 h on:12 h off dark/light cycle. The ambient animal room temperature was 70° F, controlled within $\pm 2^\circ$ and the room humidity was 50%, controlled within $\pm 5\%$. In vivo studies were not blinded, and mice were randomly assigned to treatment groups. No sample-size calculations were performed to power each study. Instead, sample sizes were determined based on prior in vivo virus challenge experiments. Mice were infused with 150 μ l of naïve or immune sera (from each immunized donor, post-boost 2 timepoint) via intraperitoneal injection. One day later, animals were intranasally inoculated with 10^3 FFU of a WA1/2020 SARS-CoV-2 strain containing the D614G mutation^{39,103}. Body weights were monitored daily. On day 6 post-challenge, the mice were sacrificed, and viral RNA was measured in the nasal turbinates, nasal wash fluid, and in the lung³⁹.

To measure the viral RNA levels, tissues were weighed and homogenized with zirconia beads in a MagNA Lyser instrument (Roche Life Science) in 1 mL of DMEM medium supplemented with 2% heat-inactivated FBS. Tissue homogenates were clarified by centrifugation at approximately 10000 g for 5 min and stored at -80°C . RNA was extracted using the MagMax mirVana Total RNA isolation kit (Thermo Fisher Scientific) on the Kingfisher Flex extraction robot (Thermo Fisher Scientific). RNA was reverse transcribed and amplified using the TaqMan RNA-toCT 1-Step Kit (Thermo Fisher Scientific). Reverse transcription was carried out at 48°C for 15 min followed by 2 min at 95°C . Amplification was accomplished over 50 cycles as follows: 95°C for 15 s and 60°C for 1 min. Copies of SARS-CoV-2 N gene RNA in samples were determined using a previously published assay¹⁰⁴. Briefly, a TaqMan assay was designed to target a highly conserved region of the N gene:

Forward primer: ATGCTGCAATCGTGCTACAA

Reverse primer: GACTGCCGCCTCTGCTC

Probe: /56-FAM/TCAAGGAAC/ZEN/AACATTGCCAA/3IABkFQ/

This region was included in an RNA standard to allow for copy number determination down to 10 copies per reaction. The reaction mixture contained final concentrations of primers and probe of 500 and 100 nM, respectively.

Edited (Page S10, line 124)

Figure S7 – Supporting information, protection following passive transfer and SARS-CoV-2 challenge. (A) Passive transfer of immune sera followed by SARS-CoV-2 D614G live virus challenge. (B) Daily body weights of the three cohorts tested: immune sera from mice immunized with DNA-VLP-30x or monomeric RBD (post-boost 2 timepoint) and sera from non-immunized C57Bl/6 as a control (n=6 mice per group, error bars represent the standard error of the mean, ***P<0.0006, ANOVA with Dunnett’s test of the area under the curve). (C-E) At day 6 post challenge tissues were harvested and viral RNA was measured in the nasal turbinates (C), nasal washes (D) and in the lung (E) (n=6 mice per group, error bars represent the standard error of the mean, ***P<0.0001, **P<0.005, *P<0.05, Kruskal-Wallis test with Dunn’s test corrected for multiple comparisons).

“2. The authors also need to demonstrate any cross-reactivity of the mice-isolated antibody sera against other viruses, preferably influenza to demonstrate the specificity of the DNA-VLP-RBS vaccine.”

Cross-reactivity to highly unrelated viruses such as influenza virus would be unprecedented. To confirm this, we evaluated reactivity of the DNA-VLP immune sera (post-boost 2 where RBD-titers are highest) to major surface antigens from two unrelated viruses: hemagglutinin (HA) from influenza virus (recombinant H1 trimer^{3,4}); and envelope glycoprotein (Env) from HIV (recombinant Clade C ZM215.8 gp120^{5,6}). Despite strong reactivity to RBD, we find no evidence of cross-reactivity to the unrelated viral spike proteins (**Reviewer 1, Figure 1**).

Reviewer 1, Figure 1. No detectable binding of antibodies in RBD-DNA-VLP immune sera (post-boost 2) to the major surface antigens on HIV (gp120 Env) or influenza virus (HA) as measured by ELISA. Upper: binding of serum IgG to RBD vs gp120 or HA. Lower: monoclonal antibody controls for binding to gp120 (VRC01), HA (CR6261) or SARS-CoV-2 RBD (B38, CR3022).

“3. The study lacks a systemic toxicity assay to determine the safety of the DNA-VLP-RBD vaccine in mice.”

We appreciate the Reviewer’s comment that the systemic toxicity of a subunit vaccine is a critical design criterion for ultimately creating a clinical vaccine product, which our current study does not address as it is beyond the scope of this pre-clinical work. However, we can point the Reviewer to two recent studies that investigated the safety profiles of DNA origami. First, we highlight our own publication that investigated the safety profile of DNA origami administered intravenously at 4 mg/kg where we observed no gross toxicity or immunotoxicity⁷. Secondly, we highlight an independent publication from the Castro group that investigated the safety profile of DNA origami administered either intravenously and intraperitoneally at doses up to 12 mg/kg and no gross toxicity or immunotoxicity were observed, including with repeated dosing regimens⁸. Given that the present study was conducted intraperitoneally and the highest dose of DNA origami in this study is 5 mg/kg, together with the preceding data in previous studies, we do not anticipate safety concerns in this system. We do note that the incorporation of protein antigens, as well as adjuvants like the Sigma adjuvant used in this work, could change the safety profile of this material system, which will be of interest to explore in future work.

Additionally, we maintain that the main contributions, in terms of novelty and impact, to the fields of immunology and vaccinology in the present study, namely that multimerizing thymus-dependent antigens on a thymus-independent scaffold does produce neutralizing antibodies against the antigen and does not produce neutralizing antibodies against the scaffold, is sufficiently demonstrated here regardless of the safety profile of the current material under investigation.

“Minor Comments:

1. The authors need to clearly mention in the abstract that the icosahedron VLPs carry the SARS-CoV-2 spike protein RBD.”

We appreciate this Reviewer’s suggestion for improving the clarity of our presentation. We have modified the Abstract to clarify this point.

Original (Page 2, line 30)

Here, we investigated DNA origami as an alternative material for multivalent antigen display in vivo, applied to the receptor binding domain (RBD) of SARS-CoV-2 that is the primary antigenic target of neutralizing antibody responses.

Edited (Page 2, line 37)

Here, we investigated DNA origami as an alternative material for multivalent antigen display in vivo, applied to the receptor binding domain (RBD) of the SARS-CoV-2 spike protein that is the primary antigenic target of neutralizing antibody responses.

“2. The authors report a T-cell-dependent antibody boosting. While not critical to the scope of the study, the author can report if the DNA-VLP-RBD vaccine also improves any T-cell responses against SARS-CoV-2, cytotoxic T-cell responses.”

We acknowledge that cellular immunity will be an important parameter to investigate in future studies. However, our focus in the present work was on humoral immunity, with our additional data set on the passive transfer of immune sera followed by live viral challenge further underscoring this focus. Nevertheless, T-cell help to enable antibody boosting was an important feature of our work showing that nanoparticle scaffolding with thymus-independent antigen did not prevent thymus-dependent memory recall/boosting of RBD antibody responses following sequential immunization with the DNA-VLP vaccine.

Reviewer #2 (Remarks to the Author in quotes and black font; Responses to the Reviewer in blue font; Revised manuscript text in red font):

Reviewer #2 (Remarks to the Author):

“445644_0

This paper has been revised and resubmitted. The text has been clarified and some new data has been added, mainly the evidence that the immune response to RBD protein presented in a multimeric form on a DNA origami scaffold is greatly reduced in TCRa $-/-$ mice. Overall, the study

shows that this multimerization approach avoids excessive antibody production directed to the multimerization moiety. A control construct with RBD multimerized on Ferritin elicited significant amounts of antibodies to the Ferritin scaffold, demonstrating that anti-scaffold responses to protein could be a problem.”

We thank the reviewer for their helpful summary of our present work, namely that multimerizing viral antigens on a DNA origami scaffold does induce neutralizing antibodies, that these antibodies are generated through a T cell-dependent pathway, and that this approach does not generate antibodies against the scaffold itself, which we and others have demonstrated is a shortcoming of protein-based antigen display platforms.

“Given that the current COVID vaccines involve mRNA delivery of minimally modified spike protein that lacks a heterologous multimerization scaffold, the approach in this paper does not really address a need, nor is it likely it would be adopted as a COVID vaccine. This materially diminishes the impact of the paper. However, there may be an argument that the vaccine approach described here would have value for vaccinations to other protein subunits of interest.”

We acknowledge the Reviewer’s assessment that, as it currently stands, our platform represents a tool for vaccine research, and nowhere in our study do we advocate for an immediate bench-to-bedside translation. However, we would push back on the premise that mRNA delivery concepts now eliminate (whole-scale) the need for basic research on particulate vaccine antigens, particularly given the decades-long successes that countless particulate vaccines have offered humanity for highly protective, long-term humoral immunity. The problem of potent-scaffold specific responses has been an outstanding question in field of nanoparticle vaccines, and we describe a previously unrecognized immunological principle that addresses this issue at the basic research level. In our view, this foundational immunological ‘trick’ we have discovered more than supplants “material diminishment” due to the very recent clinical success of mRNA-based COVID vaccines. In this respect, we strongly agree with the reviewer’s point that **“there may be an argument that the vaccine approach described here would have value for vaccinations to other protein subunits of interest.”**

“Overall, the paper is acceptable with a modest degree of enthusiasm.”

We appreciate the Reviewer’s enthusiasm for our work, and we believe that this new concept in subunit design involving the multimerization of antigens on a thymus-independent scaffold that offers antigen focusing, with proof-of-concept data in mice, will stimulate the reader’s enthusiasm for and appreciation of the novelty and impact of this work.

Reviewer #3 (Remarks to the Author in quotes and black font; Responses to the Reviewer in blue font; Revised manuscript text in red font):

“The study reports the efficacy of DNA-VLPs for its multivalent antigen presentation and negligible off-target effects. The DNA-VLPs decorated with SARS-CoV-2 Spike protein elicited a T-cell-dependent humoral response. The sera collected from immunized animals efficiently neutralized different variants of SARS-CoV-2. In summary, the authors have conducted in vivo experiments to show the vaccine efficacy of DNA-VLPs decorated with SARS-CoV-2 Spike protein. The current study is an extension of a previous report (published in Nature Nanotechnology in 2020) that is of specific interest to coronavirus research but lacks the needed depth for publication in Nature Communications. The authors need to address the following concerns before this study can be recommended for publication.”

“Major Comments:

1. The study shows efficient pseudo-virus and authentic virus neutralization with antibody sera drawn from the mice. However, the actual protection remains unexplored. The authors need to perform prophylactic/therapeutic studies to demonstrate the in vivo efficacy of the DNA-VLP-RBS vaccine.
2. The authors also need to demonstrate any cross-reactivity of the mice-isolated antibody sera against other viruses, preferably influenza to demonstrate the specificity of the DNA-VLP-RBS vaccine.
3. The study lacks a systemic toxicity assay to determine the safety of the DNA-VLP-RBD vaccine in mice.”

“Minor Comments:

1. The authors need to clearly mention in the abstract that the icosahedron VLPs carry the SARS-CoV-2 spike protein RBD.
2. The authors report a T-cell-dependent antibody boosting. While not critical to the scope of the study, the author can report if the DNA-VLP-RBD vaccine also improves any T-cell responses against SARS-CoV-2, cytotoxic T-cell responses.”

Please find our responses to these comments above in the response to Reviewer #1.

References

- 1 Hauser, B. M. *et al.* Rationally designed immunogens enable immune focusing following SARS-CoV-2 spike imprinting. *Cell Rep* **38**, 110561, doi:10.1016/j.celrep.2022.110561 (2022).
- 2 VanBlargan, L. A. *et al.* A potently neutralizing SARS-CoV-2 antibody inhibits variants of concern by utilizing unique binding residues in a highly conserved epitope. *Immunity* **54**, 2399-2416 e2396, doi:10.1016/j.immuni.2021.08.016 (2021).
- 3 Sangesland, M. *et al.* Germline-Encoded Affinity for Cognate Antigen Enables Vaccine Amplification of a Human Broadly Neutralizing Response against Influenza Virus. *Immunity* **51**, 735-749 e738, doi:10.1016/j.immuni.2019.09.001 (2019).
- 4 Sangesland, M. *et al.* Allelic polymorphism controls autoreactivity and vaccine elicitation of human broadly neutralizing antibodies against influenza virus. *Immunity*, doi:10.1016/j.immuni.2022.07.006 (2022).
- 5 Zhou, T. *et al.* Structural Repertoire of HIV-1-Neutralizing Antibodies Targeting the CD4 Supersite in 14 Donors. *Cell* **161**, 1280-1292, doi:10.1016/j.cell.2015.05.007 (2015).
- 6 Ronsard, L. *et al.* Engaging an HIV vaccine target through the acquisition of low B cell affinity. *Nat Commun* **14**, 5249, doi:10.1038/s41467-023-40918-2 (2023).
- 7 Wamhoff, E. C. *et al.* Evaluation of Nonmodified Wireframe DNA Origami for Acute Toxicity and Biodistribution in Mice. *Acs Appl Bio Mater* **6**, 1960-1969, doi:10.1021/acsabm.3c00155 (2023).
- 8 Lucas, C. R. *et al.* DNA Origami Nanostructures Elicit Dose-Dependent Immunogenicity and Are Nontoxic up to High Doses In Vivo. *Small* **18**, e2108063, doi:10.1002/smll.202108063 (2022).

REVIEWERS' COMMENTS

Reviewer #1 (Remarks to the Author):

The authors have adequately responded to the queries raised. The study is recommended for publication.

Reviewer #3 (Remarks to the Author):

The authors have adequately responded to the queries raised. The study is recommended for publication.

Reviewer #1 (Remarks to the Author in quotes and black font; Responses to the Reviewer in blue font):

Reviewer #1 (Remarks to the Author):

The authors have adequately responded to the queries raised. The study is recommended for publication.

We thank the Reviewer for their constructive and helpful peer review to improve our work.

Reviewer #3 (Remarks to the Author in quotes and black font; Responses to the Reviewer in blue font):

Reviewer #3 (Remarks to the Author):

The authors have adequately responded to the queries raised. The study is recommended for publication.

We thank the Reviewer for their constructive and helpful peer review to improve our work.